# Integration of Transcriptomics and Lipidomics Profiling to Reveal the Therapeutic Mechanism Underlying *Ramulus mori* (Sangzhi) Alkaloids for the Treatment of Liver Lipid Metabolic Disturbance in High-Fat-Diet/Streptozotocin-Induced Diabetic Mice

**DOI:** 10.3390/nu15183914

**Published:** 2023-09-08

**Authors:** Fan Wang, Sai-Jun Xu, Fan Ye, Bin Zhang, Xiao-Bo Sun

**Affiliations:** 1Institute of Medicinal Plant Development, Peking Union Medical College, Chinese Academy of Medical Sciences, Beijing 100193, China; wangfan@implad.ac.cn (F.W.); xsj365fighting@163.com (S.-J.X.); spring2378@163.com (F.Y.); 2Key Laboratory of Bioactive Substances and Resources Utilization of Chinese Herbal Medicine, Ministry of Education, Beijing 100193, China; 3Beijing Key Laboratory of Innovative Drug Discovery of Traditional Chinese Medicine (Natural Medicine) and Translational Medicine, Beijing 100193, China; 4Key Laboratory of Efficacy Evaluation of Chinese Medicine against Glyeolipid Metabolism Disorder Disease, State Administration of Traditional Chinese Medicine, Beijing 100193, China

**Keywords:** SZ-As, nonalcoholic fatty liver disease, transcriptomics analysis, lipidomics analysis, hepatic metabolome, inflammation response, lipid metabolism

## Abstract

Non-alcoholic fatty liver disease (NAFLD) is the most common liver disorder, with a global prevalence of 25%. Currently, there remains no approved therapy. *Ramulus mori* (Sangzhi) alkaloids (SZ-As), a novel natural medicine, have achieved comprehensive benefits in the treatment of type 2 diabetes; however, few studies have focused on its role in ameliorating hepatic lipid metabolic disturbance. Herein, the therapeutic effect and mechanism of SZ-As on a high-fat diet (HFD) combined with streptozotocin (STZ)-induced NAFLD mice were investigated via incorporating transcriptomics and lipidomics. SZ-As reduced body weight and hepatic lipid levels, restored pathological alternation and converted the blood biochemistry perturbations. SZ-A treatment also remarkedly inhibited lipogenesis and enhanced lipolysis, fatty acid oxidation and thermogenesis. Transcriptomics analysis confirmed that SZ-As mainly altered fatty acid oxidative metabolism and the TNF signaling pathway. SZ-As were further demonstrated to downregulate inflammatory factors and effectively ameliorate hepatic inflammation. Lipidomics analysis also suggested that SZ-As affected differential lipids including triglyceride (TG) and phosphatidylcholine (PC) expression, and the main metabolic pathways included glycerophospholipid, sphingomyelins and choline metabolism. Collectively, combined with transcriptomics and metabolomics data, it is suggested that SZ-As exert their therapeutic effect on NAFLD possibly through regulating lipid metabolism pathways (glycerophospholipid metabolism and choline metabolism) and increasing levels of PC and lysophosphatidylcholine (LPC) metabolites. This study provides the basis for more widespread clinical applications of SZ-As.

## 1. Introduction

Nonalcoholic fatty liver disease (NAFLD) is emerging as the leading chronic liver disease worldwide and afflicts approximately 25% of the adult population [1,2]. As the primer hepatic manifestation of metabolic syndrome, the rising rate of NAFLD is consistent with the rise in the global prevalence of obesity and type 2 diabetes [3]. NAFLD is usually characterized by hepatic adiposity, encompassing a range from simple steatosis to steatohepatitis; however, the disease may remain silent on the clinical manifestations until it progresses to cirrhosis [4]. Moreover, owing to its dynamics, heterogeneity and multifactorial etiology, the pathological mechanism of NAFLD is complex. These considerations make tackling NAFLD a formidable challenge, and the interventions and chemicals employed for clinical trials are also with little success. Encouragingly, armed with the advantage of high efficacy and low toxicity, natural products have increasingly gained attention as potential therapeutic agents for the treatment of NAFLD [5].

Natural products have served as a vast platform for the development of safe and cost-effective drugs or supplements. *Ramulus mori* (Sangzhi) alkaloids (SZ-As) are effective components extracted and isolated from *Morus alba* L. (mulberry twig), a traditional Chinese medicine, accounting for more than 50% of the total extract. In a double-blind randomized clinical trial, SZ-As exhibited excellent hypoglycemic effects and good safety in patients with T2D [6,7]. SZ-A tablets comprise SZ-As as major constituents and were also approved for the treatment of clinical type 2 diabetes mellitus (T2DM) in China (approval number Z20200002). Beyond their well-established glucose-lowering property, SZ-As have been confirmed to be effective in different preclinical animal models. A previous study found that SZ-As ameliorated the overall glucose metabolism and systemic inflammation in KKAy mice with signs of T2DM by improving insulin secretion and modulating the abundance of gut microbiota and gut metabolites [8]. Further study has supported the potential of SZ-As to alleviate HFD-induced obesity by regulating gut microbiota and lipid metabolism profiles [9,10]. It was also demonstrated that SZ-As protected mice from HFD-mediated hepatic steatosis, triglyceride accumulation and adipose tissue inflammation, while the anti-inflammatory role appeared to depend on the blockage of p38 MAPK, ERK and JNK signaling pathway activation in macrophages [11,12]. Notably, tissue distribution analysis suggested that after oral administration, the major constituents of SZ-As are observed predominantly in the liver, followed by the adipose tissues [13], which revealed the potential regulating effects of SZ-As on liver and adipose tissue. So far, there are relatively few reports of the role of SZ-As in treating liver lipid metabolism disorder, and their therapeutic mechanism also remains unclear.

Nowadays, multi-omics technologies focusing on quantitative and high-throughput screening are considered powerful tools for exploring the pathological disease process and identifying the molecular features behind phenotypes. Compared with single-omics analysis methods, multi-omics analysis can more systematically and comprehensively tap into the interactions and coordination mechanisms at different molecular levels of diseases through the intersection and complementarity between different omics [14,15]. The development of high-throughput sequencing, high-resolution mass spectrometry and data integration technology has promoted breakthroughs in systems biology research, as well as provided new insights and methods for target discovery and drug development of natural products characterized by multiple components and multiple targets. Recently, more and more multi-omics technologies have been applied to studying metabolic syndrome and related disorders, in order to facilitate a holistic and multidimensional understanding of metabolic diseases [16,17,18].

Taken together, this study employed the multi-omics strategy for the first time, by incorporating transcriptomics and lipidomics to investigate the mechanisms of SZ-As in ameliorating hepatic lipid metabolic disorders in NAFLD. We focus on the regulation of inflammation response based on related lipid metabolism pathways, as well as matching DEGs and differential metabolites, demonstrating the potential role of SZ-As as a natural product for the therapy of obesity and NALLD in HFD/STZ-induced mice.

## 2. Materials and Methods

### 2.1. Chemicals and Reagents

*Ramulus mori* (Sangzhi) alkaloid (SZ-A) powder (lot number: 202105002) was kindly provided by Beijing Wehand-Bio Pharmaceutical Co., Ltd., (Beijing, China). The total polyhydroxyalkaloids content in this SZ-A powder is 56.5%, which includes 1-deoxyno-jirimycin (1-DNJ) 39%, fagomine (FA) 10.5% and 1,4-dideoxy-1, 4-iminod-D-arabinitol (DAB) 7%.

### 2.2. Experimental Animals and Treatment

All animal experimental procedures were approved by the Experimental Laboratory Animal Committee of the Institute of Medicinal Plant Development, Peking Union Medical College (ethical code: Z2020002). Eight-week-old C57BL/6J mice (males) were housed at room temperature 22 °C, 60% humidity with a 12 h light–dark cycle and adequate food and water supply. After one week of adaptation, the mice were randomly divided into 4 groups: control group (Control; n = 10), model group (Model; n = 10) and SZ-A treatment by intragastrical administration group (SZ-As 50 mg/kg; n = 10, SZ-As 100 mg/kg; n = 10). NAFLD animal models were established through a single injection of 140 ug/kg streptozotocin (STZ) (Sigma-Aldrich, Saint Louis, MO, USA) four weeks after high-fat feeding. The control group was fed a standard diet, and the mice in the other groups were given a high-fat diet (HFD) (SPF Biotechnology Co., Ltd., Beijing, China) for 14 weeks. Then, the mice were treated with saline (i.g.) and SZ-As (50, 100 mg/kg, i.g.) for 8 weeks, and a high-fat diet was maintained during the treatment. Body weight was measured after 8 weeks of dosing, and glucometers (OneTouch, New Brunswick, NJ, USA) were used to determine blood glucose in the mice.

At the end of the experiment, the mice were fasted overnight before being euthanized. Blood samples were collected from the medial canthus of the mice, and the upper serum was collected for determination at 3000 rpm/15 min after standing at room temperature for stratification. The mice were disinfected with alcohol, and liver tissue, brown adipose tissue (Bat) at the scapula and inguinal white adipose tissue (iWAT) were rapidly cut from them. Liver tissues were weighed and recorded, portions of liver and adipose tissue samples were fixed in 4% paraformaldehyde and the remaining portions were placed in labeled ribozyme-free centrifuge tubes, rapidly frozen in liquid nitrogen and then transferred to −80 for storage until analysis.

### 2.3. Serum Biochemical Parameters

The levels of glycosylated protein (GSP), triglyceride (TG), total cholesterol (TC), alanine aminotransferase (ALT) and aspartate aminotransferase (AST) in mice serum were measured with a Beckman AU480 automatic biochemical analyzer according to the instructions of the testing kit (Biosano, Beijing, China). The serum levels of insulin, tumor necrosis factor-α (TNF-α), NOD-like receptor protein 3 (NLRP3), interleukin-6 (IL-6), interleukin-1β (IL-1β), interleukin-4 (IL-4) and interleukin-10 (IL-10) expression were measured using enzyme-linked immunosorbent assay kits (Sinoukbio, Beijing, China).

### 2.4. Hepatic Biochemical Indexes

Frozen liver tissues were accurately weighed and cold lysate was added, and the tissues were broken with a high-speed homogenizer to make homogenates, which were allowed to stand for 10 min and centrifuged at 4 °C, 2500 r/min for 10 min to collect the supernatants. Triglyceride (TG) content in liver tissue was measured using a biochemical kit (Nanjing Jiancheng Bioengineering Institute, Nanjing, China).

### 2.5. Histological Analysis

Liver and adipose tissues were fixed with paraformaldehyde for two days, washed and embedded in paraffin. They were deparaffinized with xylene for 5 min and washed with different gradients of ethanol and water. Sections were stained with hematoxylin dye for 5 min, rinsed with water, re-blued with a weakly alkaline aqueous solution for 30–60 s and then rinsed with water for 5–10 min. Cytoplasm was stained with eosin for 2–5 min, eluted with a gradient of ethanol, made transparent with xylene and sealed with neutral gum. Changes in liver and adipose tissue were observed with a microscope (LeicaFS1000, Wetzlar, Germany).

After the complete fixation of liver tissue, the samples were rinsed and OTC embedding medium was used to immerse the tissue, which was then frozen at −20 °C and cryostat-sectioned at 5–10 μm. Sections were infiltrated with 60% isopropanol for 1 min, stained with oil red O staining solution (Sigma-Aldrich, Saint Louis, MO, USA) for 8 min and then differentiated in 60% isopropanol. After the appropriate color was reached, the nuclei were counterstained in hematoxylin dye for 1–2 min, washed with PBS, dried and mounted with glycerin gelatin. Fat accumulation in liver tissues was observed by taking pictures with a microscope.

### 2.6. Quantitative Real-Time Polymerase Chain Reaction Analysis (RT-qPCR)

Total RNA was extracted from liver and adipose tissue using Trizol reagent (Takara Bio, Shiga, Japan), and RNA concentration was measured using a microspectrophotometer (ThermoFisher, Waltham, MA, USA). One ug of RNA was reverse-transcribed to cDNA using a reverse transcription kit (Takara Bio, Shiga, Japan), and quantitative fluorescence analysis was performed using a fluorescence quantification kit (Takara Bio, Shiga, Japan). The primer sequences are shown in Table 1.

### 2.7. RNA Preparation and RNA Sequencing Analysis

RNA extraction and sequencing were performed by Novogene Co., Ltd. (Beijing, China). The total amounts and integrity of RNA were assessed using the RNA Nano 6000 Assay Kit of the Bioanalyzer 2100 system (Agilent Technologies, Santa Clara, CA, USA) (refer to RNA quality inspection report for specific instruments). Next, mRNA was purified from total RNA by using poly-T oligo-attached magnetic beads, followed by fragmentation. In order to select cDNA fragments preferentially 370~420 bp in length, the library fragments were purified with the AMPure XP system (Beckman Coulter, Beverly, MA, USA). Then, after PCR amplification, the PCR product was purified using AMPure XP beads, and the library was finally obtained. After the library was qualified, the different libraries were pooled according to the effective concentration and the target amount of data off the machine then sequenced using the Illumina NovaSeq 6000. Finally, differential expression analysis, GO analysis, KEGG analysis and protein interaction network analysis were performed. Sequencing data quality was assessed using the GC content of nucleic acid sequences Q20 and Q30. The square of Pearson’s correlation coefficient (R^2^) was used to evaluate the repeatability and correlation between samples, and the screening of differentially expressed genes was performed after the data quality was qualified.

### 2.8. UHPLC-MS/MS Lipidomics Analysis

Collected liver tissue powder was extracted using methyl-tert-butyl ether (MTBE) solution based on the reported method [19]. We took the same amount of supernatant from each processed sample and mixed it as quality control (QC) samples. LC-MS/MS analysis was performed using a Phenomenex Kinetex UHPLC. Samples were injected onto a Thermo Accucore C18 column (2.1 × 100 mm, 1.7 μ mL) using a 16 min linear gradient at a flow rate of 0.35 mL/min. The column temperature was set at 40 °C. Mobile phase buffer A was acetonitrile/water (6/4) with 10 mM ammonium acetate and 0.1% formic acid, whereas buffer B was acetonitrile/isopropanol (1/9) with 10 mM ammonium acetate and 0.1% formic acid. The solvent gradient was set as follows: 0~10.5 min, 30% B~100% B; 10.5~12.5 min, 100% B; 12.50~12.51 min, 100% B~30% B; 12.51~16 min, 30% B. Q Exactive^TM^ series mass spectrometer was operated in positive (negative) polarity mode with sheath gas: 20 arbitrary units, sweep gas: 1 arbitrary unit, auxiliary gas rate: 5 (7), spray voltage: 3 kV, capillary temperature: 350 °C, heater temperature: 400 °C, S-Lens RF level: 50, resolving power (full scan): 120,000, scan range: 114–1700 *m*/*z*, automatic gain control target: 1 × 10^6^, resolving power (r MS 2): 30,000 (Top20), normalized collision energy: 25; 30 (20; 24; 28), injection time: 100 ms, isolation window: 1 *m*/*z*, automatic gain control target (MS^2^): 1 × 10^5^, dynamic exclusion: 15 s.

The raw data files generated through UHPLC-MS/MS were processed using the Lipid search to perform peak alignment, peak picking and quantitation for each metabolite. After that, peak intensities were normalized to the total spectral intensity. Then, peaks were matched with the Lipid search database to obtain the accurate qualitative and relative quantitative results. Statistical analyses were performed using the statistical software R (R version R-3.4.3) and Python (Python 2.7.6 version).

### 2.9. Data Processing and Multivariate Analysis

Principal components analysis (PCA) and partial least squares discriminant analysis (PLS-DA) were performed using metaX (a flexible and comprehensive software for processing metabolomics data). We applied univariate analysis (*t*-test) to calculate the statistical significance (*p*-value). The metabolites with VIP > 1, *p*-value < 0.05 and fold change ≥ 2 or FC ≤ 0.5 were considered to be differential metabolites. Volcano plots were used to filter metabolites of interest based on the log2 (FC) and −log10 (*p*-value) of metabolites. For clustering heatmaps, the data were normalized using z-scores of the intensity areas of differential metabolites and were plotted using the Pheatmap package in R language. The correlation between differential metabolites was analyzed using cor in R language (method = Pearson). Statistically significant correlations between differential metabolites were calculated using cor. mtest in in R language. A *p*-value < 0.05 was considered statistically significant, and correlation plots were plotted using the corrupt package in R language.

### 2.10. Statistical Analysis

Statistical analysis was performed using the software GraphPad Prism 9.0 for statistical analysis, and experimental data are expressed as mean ± SD. Data were subjected to a variance chi-square test, and between two groups, a two-tailed *t*-test was used; between multiple groups, one-way ANOVA was used, and two-by-two comparisons were made using Tukey’s test after the differences between the groups were statistically significant, and the differences were considered to be statistically significant when *p* < 0.05.

## 3. Results

### 3.1. SZ-As Ameliorate the Disorder of Systemic Glucose and Lipid Metabolism

Obesity and glucose lipid metabolism disorder are important manifestations of nonalcoholic fatty liver disease [20]. We first investigated the effects of SZ-As on body weight, blood glucose and lipid profile. Compared to the mice in the control group, the HFD + STZ mice in the model group had higher body weight, while the SZ-A treatment significantly improved high-glucose- and high-fat-induced obesity, in which the low-dose group (50 mg/kg) body weights were almost normalized to those of the control (Figure 1A). To further explore the role of SZ-As in metabolic regulation, we performed blood biochemical tests to detect glycemic indicators including fasting blood glucose (FBG), insulin and glycated serum protein (GSP) levels. According to the results, SZ-A treatment markedly reversed HFD/STZ-induced high serum FBG, insulin and GSP. Consistently, the levels of lipid indicators TC, TG and LDH were also lower than those of the model group (Figure 1B–E), reflecting a systemic metabolic improvement after SZ-A administration.

### 3.2. SZ-As Improve Pathological Alteration of Fatty Liver and Hepatic Function by Promoting Lipid Metabolism

Hepatic metabolic derangement could be due to the overflow of fat into the liver, which is the key feature in the development of NAFLD [21]. To further investigate the effect of SZ-As on liver steatosis caused by impaired glucose and lipid metabolism, we focused on the pathological changes of liver and adipose tissues in HFD/STZ-induced mice. Compared with the HFD + STZ group, SZ-A administration significantly decreased liver weight with either a low dose or a high dose (Figure 2A). In addition, H&E staining showed obvious hepatic vacuole degeneration accompanied by a large amount of fat deposition in the model group, exerting the distinguishing histopathological characteristic of fatty liver and liver injury. The results from oil red O staining also indicated larger clusters of red lipid droplets in hepatocytes than the control. Figure 2B shows that 50 mg/kg and 100 mg/kg SZ-As could alleviate hepatic vacuolation and hepatocyte steatosis and the 100 mg/kg dose was processed with a better effect.

Metabolic causes of fatty liver are partly genetically driven. According to previous reports, the regulation of several genes related to lipid metabolism is involved in hepatic lipid transport [22,23,24]. Thus, we examined the expression of lipogenesis and lipolysis genes like CD36, FASN, ACC, ATGL and CPT1b. As expected, the expression of lipogenesis and fatty acid transport genes was upregulated, whereas the lipolysis-related genes were decreased in mice subjected to HFD/STZ. In contrast, both doses of SZ-As were able to reverse the above effects (Figure 2C). Hepatocyte degeneration and necrosis in response to pathogenic factors leads to the release of certain enzymes from the cytoplasm and organelles into the bloodstream, resulting in altered serum concentrations of the enzymes, and these enzymes can be directly used as biomarkers in the assessment of liver injury [25]. On the other hand, the blood biochemical tests also showed elevated ALT, AST, ALP and hepatic TG levels, suggesting obvious hepatic damage and liver function abnormalities in the model group. Following the SZ-A intervention, the levels all decreased, and the differences were significant (Figure 2D–G). The above results demonstrate that SZ-As improved lipid-accumulation-induced liver impairment.

Consistent with reduced liver mass, the histologic analysis also demonstrated the multilocular lipid droplet and smaller appearance of adipocytes in SZ-A-treated iWAT and Bat (Figure 3A). Immunohistochemistry staining of UCP1 in the two kinds of adipose tissues also showed that SZ-A treatment promoted adipose browning, as indicated by much higher UCP1 expression. Further, RT-qPCR verified that SZ-As induced increased key thermogenic gene expression including Ucp1 and Cox5b in iWAT and Bat (Figure 3B,C). These data collectively illustrate that SZ-As promoted overall lipid catabolism and inhibited excess lipid synthesis, thereby reducing the hepatic burden caused by NAFLD.

### 3.3. Transcriptomics Analysis Reveals the Relevant Biological Processes and Signaling Pathways of SZ-As in the Treatment of NAFLD

To clarify the mechanism of SZ-As for NAFLD, we performed RNA sequencing (RNA-seq) analysis to systemically survey what genes were subjected to before and after SZ-A administration. By comparing the gene expression of liver tissues between the HFD/STZ group and the control group, the results indicated that there were 9658 co-modulator genes in the liver tissue of the two groups (Figure 4A). As the volcano plot represents, 4460 differentially expressed genes (DEGs) were found to be significantly expressed in the two groups, including 1848 genes upregulated and 2612 genes downregulated (Figure 4B,C). Furthermore, the gene ontology (GO) enrichment analyses revealed that the HFD/STZ-modulated genes were mainly related to the oxidoreductase activity and the fatty acid metabolic process were significantly enriched in biological processes t (Figure 4D). After SZ-A treatment, the expression of 1235 DEGs was recovered, of which 576 were downregulated and 659 were upregulated (Figure 4E,F). The GO enrichment analyses showed that SZ-A treatment significantly regulated TGF-β signaling, which has been widely reported to control collagen protein expression and subsequently impacts hepatic fibrosis [26]. We speculated that SZ-As may regulate the TGF-β pathway via the activation of downstream gene phosphorylation. As expected, the results showed that SZ-As markedly upregulated the downstream transcriptional factors targeted by TGF-β, such as smad7, smad6 and so on (Figure 4G,H). KEGG analyses further showed enrichment pathways and genes regulated by SZ-As in HFD/STZ-induced mice, including the mTOR signaling, TNF signaling and IL-7 signaling pathways (Figure 4I). Liver inflammation is one of the major features in the pathogenesis of NAFLD. Experimental evidence has suggested that TNF plays a critical role in the early events during NAFLD [27].

Based on these multiple functions of TNF in the liver, the heatmap shows information on the transcript-level changes among the control, model and ST-Z administration groups, maintaining a marked “low-high-low” trend of the three groups (Figure 5A). We further speculated that TNF affected the expression of lipid-metabolism-related genes such as CEBPα and AP-1 by modulating the transcription of inflammatory factors, ultimately achieving the crosstalk between hepatic inflammation and metabolism. (Figure 5B). To validate the above findings, inflammation-associated factors were analyzed. The levels of pro-inflammatory factors NLRP3, TNF-α, IL-6 and IL-1β were significantly increased compared to the control group, while SZ-A treatment downregulated these factors’ levels compared with those of the model group (Figure 5C–F). Furthermore, HFD/STZ induced downregulation of the anti-inflammation factors IL-4 and IL-10, and there also was obvious upregulation after SZ-A administration (Figure 5G,H). These observations from RNA-seq analysis reveal that TNF-mediated inflammation is involved in the SZ-A treatment of NAFLD.

### 3.4. Lipidomics Analysis Reveals SZ-As Reduced the Hepatic Metabolic Level of Triacylglycerols (TGs) and Phosphatidylcholines (PCs) in HFD/STZ-Induced Mice

To investigate the metabolic profile of the HFD/STZ model and mice administered SZ-As, LC-MS/MS was used to acquire lipidomics data for the liver tissue in each group. The PCA score plot indicates the separated metabolic states of the SZ-A group and model groups, revealing there was a good difference in the metabolites of the two groups (Figure 6A). Differentially regulated representative metabolites are illustrated with the volcano plot using the parameters VIP > 2.5 and *p* < 0.05 (Figure 6B). Through analysis of the common differential metabolites between the metabolites selected from the control vs. model group and the model group vs. SZ-A group, 28 candidate metabolites were screened out from the control, model and administration groups, such as free fatty acid (FFA), lysophosphatidylcholine (LPC), triacylglycerols (TGs), phosphatidylcholines (PCs) and so on (Figure 6C,D). Heatmap visualization (Figure 6E) also showed that 28 metabolites were distinctly restored following SZ-A administration. Among them, SZ-As may affect the synthesis of the two differential metabolites, PCs and TGs, ultimately improving the disorder of hepatic lipid metabolism (Figure 6F). To further delineate the underlying mechanism of the differential metabolites, we performed metabolic pathway and related target analysis. The main metabolic pathways determined via KEGG analyses in the SZ-A and model groups included the adipocytokine signaling pathway, glycerophospholipid metabolism, choline metabolism and the sphingolipid signaling pathway (Figure 6G). These results demonstrated that SZ-A treatment, to a certain degree, reversed the HFD/STZ-induced metabolite disorder. Therefore, it is speculated that the role of SZ-As may be related to lipid metabolism pathways via controlling the synthesis of PCs and TGs, and to the goal of treating NAFLD.

### 3.5. Integrated Analysis of Lipidomics and Transcriptomic Data of SZ-A-Treated Mice

In order to explore the specific metabolic pathway of SZ-As in the liver of HFD/STZ mice, the biological differential metabolites and DEGs were imported into Cytoscape for joint-pathway analysis. According to KEGG, with the integration of DEGs and lipidomics data, 11 metabolism-related common pathways were identified including butanoate metabolism, linoleic acid metabolism, glycerophospholipid metabolism, glutathione metabolism, and other pathways. Previous related research has reported that glycerophospholipid and choline metabolism are significant for the pathogenesis of NAFLD function; therefore, we speculated that these metabolic pathways may be the key process of SZ-As in the treatment of NAFLD (Figure 7A,B). Meanwhile, to better capture the direct changes of DEGs in the above core metabolic pathways, a heatmap was constructed to give a pictorial view (Figure 7C,D). Among the glycerophospholipid- and choline-metabolism-related metabolites, SZ-As significantly increased the levels of candidate metabolite PCs and LPC, indicating that SZ-As were correlated with protection against glycerophospholipid metabolic disorder (Figure 7E). Finally, Figure 7F summarizes the effects of SZ-As on NAFLD in the network of screened metabolic biomarkers and DEGs. These observations, which were based on the core genes and key metabolites, suggested the role of SZ-As in treating HFD/STZ-induced liver lipid metabolic disturbance.

## 4. Discussion

Non-alcoholic fatty liver disease (NAFLD) is a heterogenous chronic disease closely associated with metabolic irregularities [28]. Ectopic fat deposition and persistent inflammation caused by metabolic disturbance of hepatocytes is the hallmark of NAFLD, in which abnormalities of metabolites and signal pathways might play more important roles in the development of the disease [29,30]. SZ-As, derived from Morus alba L, have exerted multiple pharmacological effects in terms of lowering hyperglycemia and improving obesity and systemic inflammation [31]. Given the highly similar pathogenesis shared between NAFLD, obesity and T2DM [32], we focus on the therapeutic effects of SZ-As on NAFLD. At present, research on the role of SZ-As in obesity-associated liver injury is still lacking, especially in terms of molecular mechanisms.

In this study, the NAFLD model in mice was established using HFD (14 weeks), combined with streptozotocin (HFD/STZ), and then the action of SZ-As against fatty liver was validated following administration of SZ-As for another 8 weeks. An uncontrolled increase in body mass index is an important pathogenic factor for NAFLD [33]. We found that both the low dose and the high dose of SZ-As significantly reduced HFD/STZ-induced obesity. It is reported that the inhibition of insulin-induced muscle glucose transport and glycogen synthesis following high fat and high sugar intake is thought to occur first, thereby resulting in the redirection of undisposed excess glucose to the liver and promotion of hepatic de novo lipogenesis (DNL), which ultimately lead to the occurrence of NAFLD [34,35]. Consistent with this assertion, daily supplementation with SZ-As decreased the sugar metabolism measures including the glycemic index and insulin and GSP content in serum. Furthermore, the glucose-converted DNL is almost metabolized to triglycerides (TGs) in the liver and further causes hypertriglyceridemia [36]. We also found decreased TG and TC levels in the serum and liver after SZ-A administration. These findings demonstrated that the NAFLD model was effectively established in the HFD/STZ-treated mice, in which the SZ-As markedly improved systemic metabolic dysfunction.

In order to further clarify the effect of SZ-As on liver lipid metabolism induced by HFD/STZ, we performed liver and adipose tissue phenotype validation. Compared with the model group, the liver weight of mice and the size of lipid droplets in hepatocytes were remarkedly reduced in the SZ-A treatment group, which indicated decreased ectopic lipid deposition in the livers of the mice. Fatty acid accumulated in the liver may lead to damage to liver cells, triggering an inflammatory response, in turn resulting in elevated transaminase levels [37]. High-fat diet and STZ injection could induce severe hepatic impairment, elevating serum ALT, AST, ALP and LDH levels, which were reduced with SZ-A treatment, suggesting a hepatoprotective effect of SZ-As. The metabolic causes of NAFLD are most often genetically regulated. Based on the auto-activation of hepatic lipogenesis and lipid spillover from adipose tissues to the liver in the development of NAFLD [38,39], we found that after SZ-A intervention, key genes related to hepatic lipid synthesis, such as FASN and ACC, were downregulated. CD36 and CPT-1b are both key enzymes involved in fatty acid transport and oxidative catabolism. CD36 serves as a receptor and transporter carrier for long-chain fatty acid (LCFA), facilitating the shuttling and translocation of LCFA in adipocytes and hepatocytes. CPT-1b is responsible for catalyzing hepatic mitochondrial β-oxidation to ameliorate hepatic steatosis [40,41,42]. In our study, the CD36 level in the liver was significantly elevated in the model group, and its expression was inhibited by SZ-As, while the level of CPT-1b showed the opposite trend. Moreover, the expression of ATGL, a representative gene for lipolysis, was upregulated by SZ-As. Through examining the target genes of various pathways related to hepatic lipid synthesis and catabolism, fatty acid transport and oxidation in experimental mice, we believe that SZ-As could play a hypolipidemic role by decreasing hepatic lipid synthesis, regulating lipid oxidation metabolism and promoting fat catabolism. Adipocyte dysfunction also affects NAFLD [43,44]. Consistent with this result, histological analysis of adipose indicated SZ-As contributed to reducing lipid droplet size and increasing thermogenesis protein Ucp1 expression. The levels of lipid-metabolism-related regulatory factors such as Ucp1 and Cox5b were also enhanced by SZ-As. These data confirm a favorable therapeutic effect on hepatic lipid metabolism disorders in NAFLD.

Numerous studies are employing multi-omics techniques to understand the underlying therapeutic mechanisms of natural products systematically and comprehensively [45]. Here, to determine the mechanism by which SZ-As alleviate fatty liver, we first established the gene expression profiles of the control vs. model group and model vs. SZ-A group using RNA-Seq analysis. GO enrichment analysis of the differential genes revealed that cholesterol metabolism, acetyl coenzyme A metabolism and fatty acid metabolism were associated with HFD/STZ-induced steatohepatitis, indicating that HFD/STZ mice might have disorders of lipid metabolism. In metabolic-dysfunction-associated NALFD, inflammatory processes are important triggers that drive advanced fibrosis [46]. Our study found that cellular functions were enriched for TGF-β signaling after SZ-A administration. TGF-β is one of the inflammatory mediators and has been demonstrated as a prognosis biomarker for assessing the risk of hepatic steatosis and fibrosis development [30,47]. In addition, there is a significant correlation between TGF-β1/Smad3 signaling and adiposity [48,49]. TGF-β was also shown to be involved in the regulation of adipogenic differentiation, which is mediated by the inhibition of PI3K/Akt activation on the Smad pathway [50,51]. This suggests that SZ-As altered the processes of lipid metabolism disturbance and inflammatory response. KEGG analysis further indicated that these genes were mainly enriched in insulin resistance, inflammation-related and adipocytokine signaling pathways.

Previous studies showed that the anti-obesity and lipid-lowing properties of SZ-As were attributed to ameliorating adipose inflammation [8,11]. TNF is recognized as a classical regulator that synergistically mediates signaling to drive inflammation and is critical for the advancement of obesity [52,53]. In the liver, TNF induces numerous biological responses such as insulin resistance, liver inflammation and hepatocyte apoptosis and necroptosis, which further interconnects inflammation with metabolic signals [27,54,55]. In line with these studies, we screened for genes closely associated with the TNF pathway among differential genes and examined the level of cytokines related to this pathway. SZ-As were observed to tend to reduce the levels of pro-inflammatory factors and promote the production of anti-inflammatory factors. Impaired lipid metabolism in the liver could re-inflame inflammatory responses at tissue sites, and the elevation of inflammatory factors further destroys the hepatic biological functions, which illustrates the interactions between aberrant lipid metabolism and inflammation [56]. Accordingly, it suggests that the protective effect of SZ-As on the fatty liver may be related to the inhibition of inflammatory injury by regulating lipid metabolism.

Overaccumulated lipids can be oxidized, esterified and restored in the liver, leading to metabolite leakage such as phospholipids, triglycerides, fatty acids and so on [57]. In order to further explore the regulation of lipid metabolism and inflammation process by SZ-As, an untargeted lipidomics study was performed. The major metabolic pathways identified at the metabolomic level in the SZ-A and the model group included glycerophospholipid, sphingomyelin, choline and multiple lipid metabolism. Perturbations of lipid metabolism are responsible for the development of NAFLD. Hydrolyzed lipids enter the hepatocytes in the form of free fatty acids and are either β-oxidized to produce energy or stored in the liver as TG. Moreover, hepatocytes use free fatty acids or glucose for the biosynthesis of glycosphingolipids, which are principal components of cell membranes and play important roles in intracellular signal transduction [58,59]. PCs and LPC are key byproducts in the glycerophospholipid metabolism pathway [60]. The liver appears to be critically dependent on PC synthesis, especially in overnutrition, where a decreased PC level is implicated in the development of NAFLD. In the latest study, LPCs were found to drive hepatic PC formation and suppress lipogenesis to contribute to sustaining the hepatic pool processes that are essential to protect the liver from excessive dietary fat [61]. Consistent with this, our analysis revealed that the relevant metabolites TGs, PCs and LPC were significantly disturbed before and after SZ-A administration. Significantly increased levels of PCs and LPC were observed after SZ-A treatment compared with those in model mice. Interestingly, several findings showed that the exogenous PCs could promote the polarization of M2-type macrophages and inhibit p65 NF-κB phosphorylation, thereby attenuating chronic inflammation to ameliorate obesity and related disorders [62,63]. These discoveries build a network between inflammatory response with PC synthesis, although no direct alternation in inflammatory genes has been found in the integrated transcriptome and liposome analysis. It is hypothesized that SZ-As may control the PC level to further influence inflammatory reaction in protein dimensions. Finally, we concluded that SZ-As regulating the genes involved in glycerophospholipid and choline metabolism, such as Lpcat, Lypla and Pnpla7, as well as the abundance of LPC and PC metabolites, ultimately alleviates HFD/STZ-induced NAFLD.

Our current work presents some limitations that will be the focus of future research. The mechanism of regulation is not clear, though we found that the lipid metabolism process was influenced by SZ-A administration in NAFLD mice. Moreover, the effect of SZ-As on hepatic cells in vitro awaits further investigation. Considering the integrated transcriptome and lipidomics analysis, candidate target genes for SZ-As appear to have been screened; however, further study will be necessary to verify the discrepancy of these annotations. It will be interesting to explore the roles of these key metabolites as well as the related target genes in the progression of NAFLD.

## 5. Conclusions

SZ-As have significant anti-glucose and anti-obesity effects. However, there are few studies on the impact on non-alcoholic fatty liver disease (NAFLD) of SZ-As, and the mechanism is not clear. In this study, we observed SZ-A regulation of hepatic lipid metabolism disorder in HFD/STZ-induced NAFLD mice and further explored this mechanism using integrated transcriptome and lipidomics analysis. The results suggested that SZ-As could alleviate inflammation and save damaged lipid metabolism by modulating the TNF signal pathway, which is related to the regulation of glycerolphospholipid and choline metabolism. These findings suggest that SZ-As are a promising drug for the treatment of NAFLD.

## Figures and Tables

**Figure 1 nutrients-15-03914-f001:**
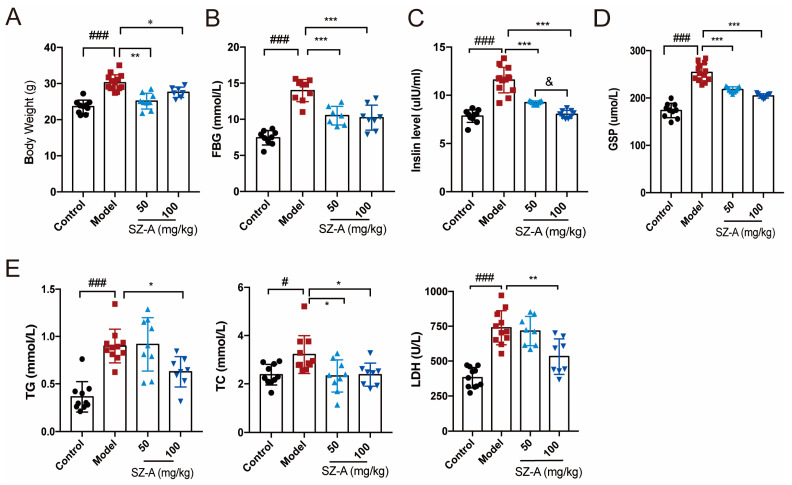
Effect of SZ-As on systemic metabolic function of HFD/STZ-induced diabetic mice. (**A**) Body weight. (**B**–**E**) Serum biochemical indexes of mice. The data represent the mean ± SD. n = 10, # *p* < 0.05, ### *p* < 0.001 versus the control; * *p* < 0.05, ** *p* < 0.01, *** *p* < 0.001 versus the HFD + STZ; & *p* < 0.05 versus the 50 mg/kg SZ-As group.

**Figure 2 nutrients-15-03914-f002:**
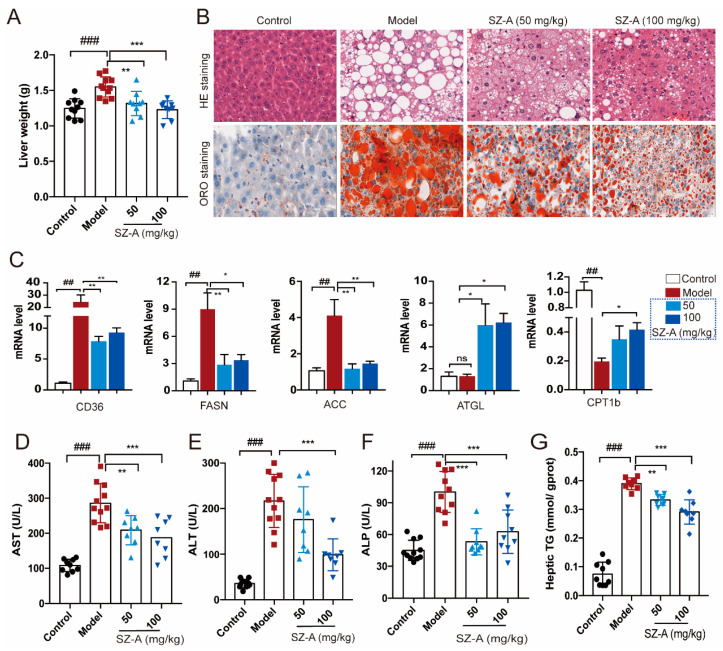
Effects of SZ-As on liver function and hepatic lipid levels in mice. (**A**) Liver tissue weight. (**B**) Representative images of hematoxylin–eosin (H&E)- and oil red O-stained sections of liver tissues (scale bar, 200 μm). (**C**) RT-qPCR analysis of CD36, FASN, ACC, ATGL and CPT 1b expression in livers (n = 6/group). Effects of SZ-As on serum AST (**D**), ALT (**E**), ALP (**F**) and (**G**) TG levels of the liver in the groups of mice (n = 8 or 10/group). The data represent the means ± SD. ## *p* < 0.01, ### *p* < 0.001 versus the control; * *p* < 0.05, ** *p* < 0.01, *** *p* < 0.001 versus the HFD + STZ.

**Figure 3 nutrients-15-03914-f003:**
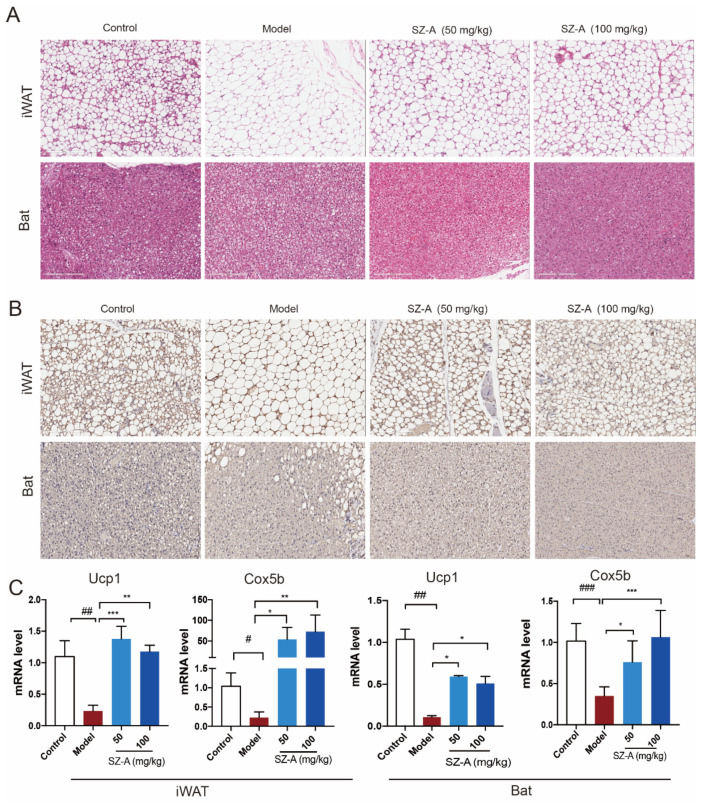
Adipose metabolic profiles and thermogenesis levels of HFD/STZ-induced mice after SZ-A treatment. Representative hematoxylin and eosin (H&E) staining images (**A**) and Ucp1 immunohistochemical staining images (**B**) of mouse iWAT and Bat, scale bar: 200 μm. (**C**) RT-qPCR analysis of Ucp1 and Cox5b expression in iWAT and Bat (*n* = 6/group). The data represent the means ± SD. # *p* < 0.05, ## *p* < 0.01, ### *p* < 0.001 versus the control; * *p* < 0.05, ** *p* < 0.01, *** *p* < 0.001 versus the HFD + STZ.

**Figure 4 nutrients-15-03914-f004:**
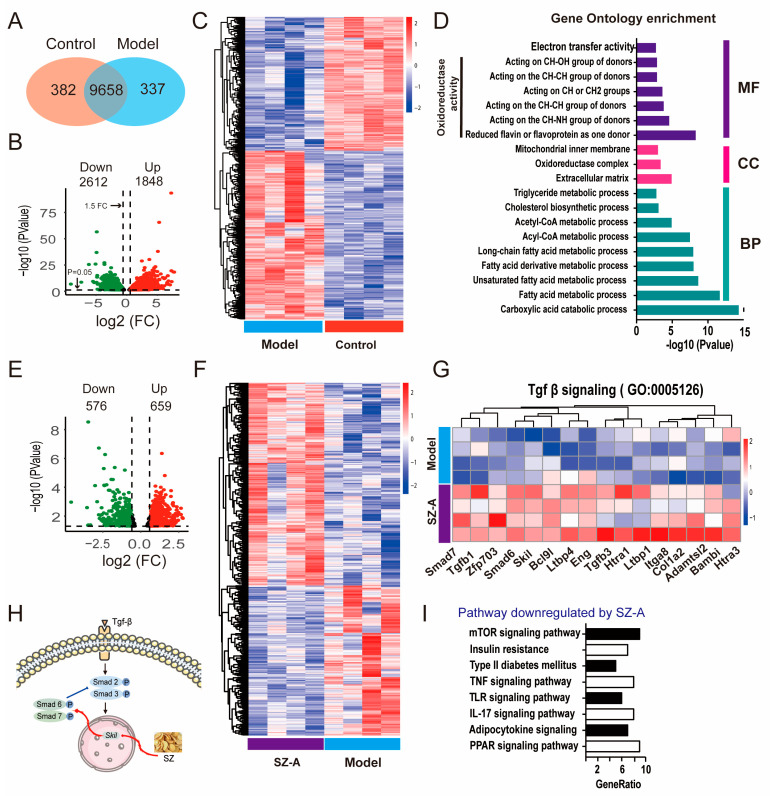
Transcriptomics analysis results of SZ-A-treated mice with HFD/STZ. (**A**) Venn diagram of RNA-seq indicates the number of co-expressed genes between control and model mice, respectively (n = 3). (**B**) Volcano plot of up- and downregulated DEGs between the control and model groups (*p* < 0.05 and |log2FoldChange| > 1.5). The green points represent the downregulated genes; the red points represent the upregulated genes. (**C**) Heatmap of RNA-seq between the control and model groups. (**D**) GO enrichment bar plot for DEGs between the control and model groups. (**E**) Volcano plot of up- and downregulated DEGs between the model and SZ-A groups (*p* < 0.05 and |log2FoldChange| > 1). The green points represent the downregulated genes; the red points represent the upregulated genes. (**F**) Heatmap of RNA-seq between the model and SZ-A groups. (**G**) Heatmap of Tgf-β signaling between the model and SZ-A groups. Red and blue represent up- and downregulation expression, respectively. (**H**) Diagrammatic sketch of SZ-As acting on Tgf-β-Smad signaling pathways. (**I**) KEGG analysis of the downregulated pathways between the model and SZ-A groups.

**Figure 5 nutrients-15-03914-f005:**
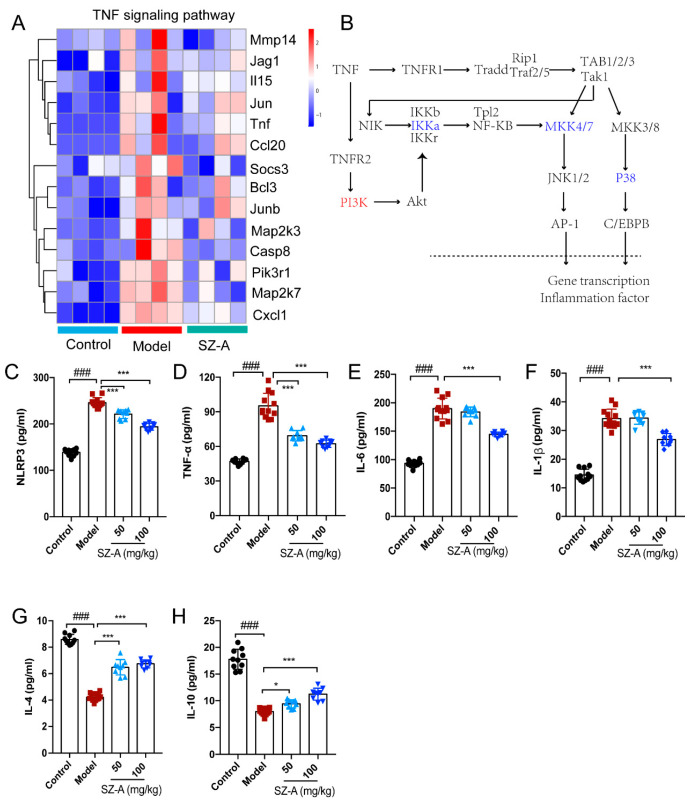
Effect of SZ-A treatment on tumor necrosis factor (TNF) pathway in mice. (**A**) Heatmap of TNF pathway gene clusters among the control, model and SZ-A groups. (**B**) Diagrammatic sketch of the regulation of the genes in the TNF pathway. Inflammation-related serum NLRP3 (**C**), TNF-α (**D**), IL-6 (**E**) IL-1β (**F**), IL-4 (**G**) and IL-10 (**H**) expression levels in mice. The data represent the mean ± SD. N = 10, ### *p* < 0.001 versus the control; * *p* < 0.05, *** *p* < 0.001 versus the HFD + STZ.

**Figure 6 nutrients-15-03914-f006:**
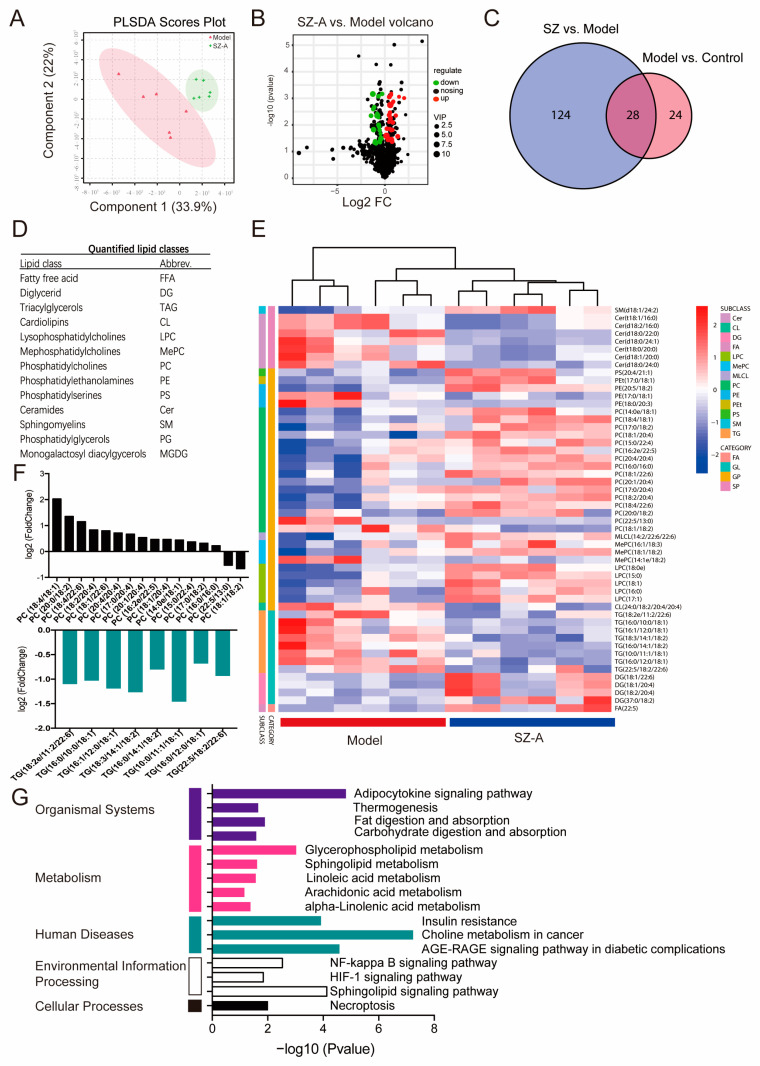
Lipidomics analysis of liver tissue treated with SZ-As. PCA score plot (**A**) and volcano plot (**B**) between the model and SZ-A groups. (**C**) Venn diagram of the number of differential metabolites among groups. (**D**) Differentially expressed lipid classes. (**E**) Heatmap visualization of the intensities of metabolic biomarkers between the model and SZ-A groups. (**F**) Bar plot of differentially expressed lipids between the model and SZ-A groups. (**G**) Pathway analysis of differential metabolites.

**Figure 7 nutrients-15-03914-f007:**
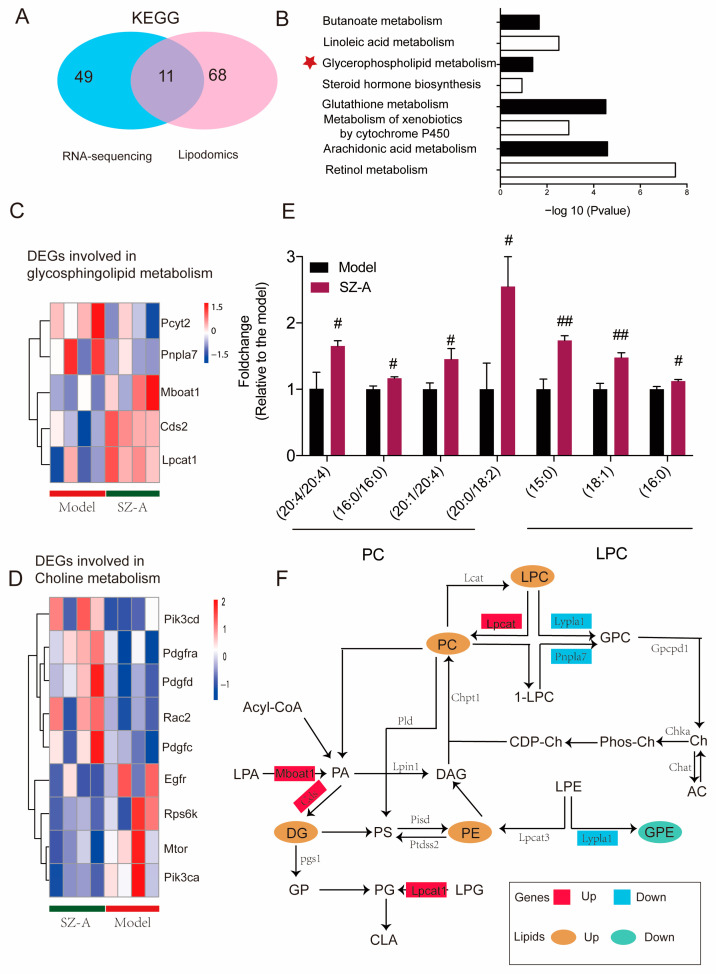
Integrated analysis of lipidomics and transcriptomics studies. (**A**) Venn diagram of the number of KEGG differential metabolite pathways among groups. (**B**) Bar plot of KEGG differential metabolite pathways among groups. Heatmap of key DEGs enriched in glycerophospholipid (**C**) and (**D**) choline metabolism gene clusters between the model and SZ-A groups. (**E**) Bar plot of representative differentially expressed lipids between the model and SZ-A groups. (**F**) The comprehensive approach of differential genes regulating differential lipids. The data represent the mean ± SD. N = 6, # *p* < 0.05, ## *p* < 0.01 versus the model.

**Table 1 nutrients-15-03914-t001:** The primer sequences.

Gene	Forward Sequence (5′–3′)	Reverse Sequence (3′–5′)
CD36	ATGGGCTGTGATCGGAACTG	ATGGGCTGTGATCGGAACTG
FASN	GGAGGTGGTGATAGCCGGTAT	TGGGTAATCCATAGAGCCCAG
ACC	GATGAACCATCTCCGTTGGC	GACCCAATTATGAATCGGGAGTG
ATGL	CAGGAGTTGATTCCAGACAGGTA	CAGGAGTTGATTCCAGACAGGTA
CPT1b	GCACACCAGGCAGTAGCTTT	CAGGAGTTGATTCCAGACAGGTA
Ucp1	CAGGAGTTGATTCCAGACAGGTA	CAGGAGTTGATTCCAGACAGGTA
Cox5b	CAGGAGTTGATTCCAGACAGGTA	CAGGAGTTGATTCCAGACAGGTA

CD36, platelet glycoprotein 4; FASN, fatty acid synthase; ACC, 1-aminocyclopropane-1-carboxylicacid; ATGL, adipose triglyceride lipase; CPT1b, carnitine palmitoyltransferase 1b; Ucp1, uncoupling protein 1; Cox5b, cytochrome c oxidase subunit 5b.

## Data Availability

The data of this study are available on request from the corresponding author.

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
