# Peer review of "Integration of Transcriptomics and Lipidomics Profiling to Reveal the Therapeutic Mechanism Underlying *Ramulus mori* (Sangzhi) Alkaloids for the Treatment of Liver Lipid Metabolic Disturbance in High-Fat-Diet/Streptozotocin-Induced Diabetic Mice"

_nutrients, 2023, doi:10.3390/nu15183914_

Round 1

Reviewer 1 Report

In this manuscript, the authors explored the therapeutic potential of Ramulus Mori (Sangzhi) Alkaloids (SZ-A) for diabetes.  The authors found that SZ-A improved metabolic syndrome in HFD/STZ-induced diabetic mouse model.  The authors then integrated transcriptomics and lipidomics approaches and identified several pathways that may be involved in the effects of SZ-A.  This study provides a significant amount of data and the manuscript is well prepared in general.  The results are informative and provide several directions for their future studies.

Author Response

Many thanks for your kind comments on our manuscript. We appreciate your efforts in reviewing our manuscript. Your encouraging review has helped us build self-confidence and self-worth. Once again, thank you very much for your comments and suggestions. 

Reviewer 2 Report

In the present study, integration analysis of transcriptomics and lipidomic was applied to reveal the therapeutic mechanism of liver lipid metabolic disturbance in HFD/STZ-induced diabetic mice treated by Sangzhi alkaloids, which is a very meaningful research work. Biochemical indicators related to glucose and lipid metabolism, liver and inflammation was analyzed. Moreover, Individual and integrated analysis for transcriptomics and lipidomic were also conducted.  However, there existed some problems that this paper cannot be published in present status. My individual comments were listed below.

1. In the part of Chemical and Reagents, the description of SZ-A composition is not appropriate. DNJ, FA and DAB should be expressed by full name.

2. The description of Figure 3 A and B was too simplemore detailed information in the text should be added.

3. The content of the discussion section is more like introduction. The author should rewrite and focus on discussing the content related to the results.

4. The conclusion of integrated analysis of lipidomic and transcriptomic data in abstract part is inaccurate, especially for the TNF signaling pathway, which was not mentioned in the part of “Integrated Analysis of Lipidomics and Transcriptomic Data of SZ-A- treated Mice”.

Author Response

Many thanks for your comments and kind advice on our manuscript. We have studied these comments carefully, answered them point-by-point, and made modifications and corrections that we hope them meet your approval. All these suggestions have improved our manuscript and broadened our knowledge.

Point 1: In the part of Chemical and Reagents, the description of SZ-A composition is not appropriate. DNJ, FA, and DAB should be expressed by full name.

Response 1: We are grateful for the suggestion. According to your comment, we have added the full name of DNJ, FA, and DAB to the Chemical and Reagents section of the manuscript. (Lines 93-97, pages 2-3)

Point 2: The description of Figure 3 A and B was too simple, more detailed information in the text should be added.

Response 2: We agree with the comment. We have revised the text to address your concerns and hope that it is now clearer. (Lines 280-285, page 7)

Point 3: The content of the discussion section is more like introduction. The author should rewrite and focus on discussing the content related to the results.

Response 3: Thank you for your suggestions. We agree with you that the description of the discussion section lacks clarity, and therefore substantial modification has been made. Concretely, we simplified the description of something similar to introduction and made a more concise statement of the results in the content. We also added some critical assessments of the results as well as future considerations. In addition, a few spaces have been added to highlight the limitations of the research. (Lines 405-530, pages 14-17)

Point 4: The conclusion of integrated analysis of lipidomic and transcriptomic data in abstract part is inaccurate, especially for the TNF signaling pathway, which was not mentioned in the part of “Integrated Analysis of Lipidomics and Transcriptomic Data of SZ-A- treated Mice”.

Response 4: Thank you very much for pointing out this problem. We have rewritten the corresponding abstract section. (Lines 29-33, page 1)

We tried our best to improve the manuscript and made some changes in the manuscript. These changes will not influence the content and framework of the paper. And here we did not list the changes but marked them in red in the revised paper.

Thank you for your careful review once again. We appreciate your efforts in reviewing our manuscript. Your careful review has helped to make our manuscript clearer and more comprehensive.

Reviewer 3 Report

Many plants or plant extracts are used in traditional Chinese medicine. The reviewed article presents research on the use of alkaloids isolated from Morus alba L. (SZ-A). The introduction to the article was written correctly based on the latest scientific literature. Abstract raises some doubts. It should primarily be a summary of the results written in a clear way and encouraging the reader to read the entire article. General information such as at the beginning of the Abstract is redundant in it. Therefore, in my opinion, the authors should improve Abstract.

The research described in the article was planned and conducted by the authors in a logical way. The results are presented clearly, which facilitates their analysis. However, in my opinion, the authors should address two issues.

1.       In section 2.8, the authors describe the UHPLC separation conditions they used. However, it is difficult to find a detailed reference to their results in the following text. There is no information on how the sample for injection was prepared, what substances the analysis concerned and how individual components were identified (molecular ion, fragmentation).

2.       Even more doubtful is the GC-MS analysis mentioned in line 342. Apart from this point, there is no reference to this analysis throughout the text. There is no description of the analysis conditions. There is no information about the preparation of the sample, especially whether it required derivatization.

In the case of such a general and brief description, it is difficult to assess whether the analyzes were carried out correctly, and even more so to assess the correctness of any results. Therefore, in my opinion, the authors should complete and correct this description.

Author Response

Thank you for your precious comments and advice. Those comments are all valuable and very helpful for revising and improving our paper. We have made corrections accordingly which we hope to meet your approval. Revised portions are marked in red on the paper and our point-by-point responses are detailed below.

Point 1: Abstract raises some doubts. It should primarily be a summary of the results written in a clear way and encouraging the reader to read the entire article. General information such as at the beginning of the Abstract is redundant in it. Therefore, in my opinion, the authors should improve Abstract.

Response 1: We are grateful for your suggestions. According to your comments, we have simplified the description at the beginning of the abstract and embellished the content appropriately based on the full text. (Lines 16-20, pages 1)

Point 2: In section 2.8, the authors describe the UHPLC separation conditions they used. However, it is difficult to find a detailed reference to their results in the following text. There is no information on how the sample for injection was prepared, what substances the analysis concerned and how individual components were identified (molecular ion, fragmentation).

Response 2: Many thanks for your valuable comments. We agree with you that there is indeed something ambiguities in this section, and therefore we rewrote this part. In addition to retaining the contents of chromatographic and mass spectrometry conditions, the methods of how to extract metabolites to obtain quality control (QC) samples, as well as qualitative and quantitative analysis processes of lipid data were added. Notably, the content corresponding to section 2.8 is in section 3.4 of the results, in which GC-MS is revised to LC-MS. (Lines 182-204, pages 4-5)

Point 3: Even more doubtful is the GC-MS analysis mentioned in line 342. Apart from this point, there is no reference to this analysis throughout the text. There is no description of the analysis conditions. There is no information about the preparation of the sample, especially whether it required derivatization. In the case of such a general and brief description, it is difficult to assess whether the analyzes were carried out correctly, and even more so to assess the correctness of any results. Therefore, in my opinion, the authors should complete and correct this description.

Response 3: We appreciate for pointing out this problem and apologize for our error. Corresponding to the method mentioned in 2.8, LC-MS not GC-MS was performed to detect and analyze lipid molecules in liver tissue. We have revised this description. (Line 352, page 11)

We sincerely hope that this revised manuscript has addressed all your comments and suggestions. We appreciate your careful review and warm work earnestly and all these suggestions have improved our manuscript and broadened our knowledge. Once again, thank you very much for your comments and suggestions.